# Psychometric Properties of General Self-Efficacy (GSE) Scale Korean Version for Older Korean Immigrants with Diabetes: A Cross-Sectional Study in the United States

**Jung Eun Kim** [1,*][iD]**, Ying-Hong Jiang** [2] **and Vivien Dee** [3]

[1]   Mennonite College of Nursing, Illinois State University, Normal, IL 61790, USA
[2]   School of Education, Azusa Pacific University, Azusa, CA 91702, USA; yjiang@apu.edu
[3]   School of Nursing, Azusa Pacific University, Azusa, CA 91702, USA; vdee@apu.edu
**\***   Correspondence: jkim133@ilstu.edu

**Abstract:** Patients with diabetes must have self-efficacy to perform necessary self-care tasks. Self-efficacy has been considered as one of the primary motivators on diabetes self-care; therefore, it is essential for health care professionals to assess the self-efficacy of patients with diabetes to provide optimal care. Despite older Korean immigrants having greater difficulty in diabetes management, research on self-efficacy for them is lacking. This study aims to examine the psychometric property of the General Self-Efficacy scale Korean version for older Korean immigrants with diabetes in the United States. In this cross-sectional, methodological study, data were collected using convenience sampling. Cronbach's alpha, exploratory factor analysis, and confirmatory factor analysis were employed to examine the psychometric properties. Cronbach's alpha for the entire GSE scale Korean version is 0.81. The initial Eigenvalues show two factors, coping and confidence; however, the confirmatory factor analysis showed reasonable goodness of fit to the data ($\chi^2(35) = 86.24$, $p < 0.01$), $\chi^2/df$ ratio = 2.46, AGFI = 0.87, GFI = 0.91, IFI = 0.90, ECVI = 0.74, CFI = 0.89, and RMSEA = 0.093 in the one-factor model. The General Self-Efficacy scale Korean version demonstrated acceptable reliability and validity. It can be used to investigate self-efficacy and to devise culturally tailored diabetes interventions.

**Keywords:** self efficacy; psychometric; diabetes mellitus; emigrants and immigrants; reliability; validity

## 1. Introduction

In 2021, diabetes affected approximately 537 million adults (20 to 79 years) worldwide, with type 2 diabetes (T2D) accounting for approximately 90% of all types of diabetes [1]. Particularly, diabetes is widely acknowledged as a prevalent chronic condition that significantly affects the morbidity and mortality of older people. Approximately 48.8% of adults 65 years or older (26.4 million) have prediabetes in the United States [2]. Due to aging-related barriers, older individuals with diabetes face more challenges in maintaining optimal health status. Even though self-care is the most essential part of the effective management of diabetes, many older adults have difficulty performing effective self-care, because performing daily self-care activities is not simple. It is complex and includes multiple tasks such as diet changes, medication taking, monitoring blood sugar, and regular medical visits [3]. If patients with diabetes do not carry out the required self-care tasks correctly, they may experience acute or long-term consequences, such as eye and skin complications, functional disorder, neuropathy, hypertensive disorder, stroke, and even mortality [4].

To initiate self-care activities, individuals with diabetes must have internal motivation. Self-efficacy has been considered one of the primary motivators to change behaviors [5]. Bandura first proposed the concept of self-efficacy [5]; it represents an individual's belief

in their capacity to perform particular practices or tasks. Self-efficacy influences the efforts individuals are willing to exert in the face of barriers, obstacles, or failures [6]. For individuals with chronic illnesses, self-efficacy is the belief that one can exert control over challenging circumstances [7]. In other words, people with high self-efficacy are more likely to possess the highest levels of behavioral change ability [8]. In terms of behavior change, diabetes requires behavior changes to healthy lifestyles, and individuals with diabetes should have sufficient self-efficacy. Much of the previous literature reported that self-efficacy is one of the most important factors in self-care in patients with diabetes [8–10]. Therefore, measuring and identifying levels of self-efficacy for people with chronic disease such as T2D are essential for health care professionals to provide customized, effective health care services.

In 1981, Matthias Jerusalem and Ralf Schwarzer established the General Self-Efficacy (GSE) scale; the scale was created to assess a person's general sense of self-efficacy and self-beliefs in coping with daily challenges and adapting to all types of stressful life events [11]. The original developers of the GSE scale granted permission to other researchers to reproduce or employ it in future studies. The GSE scale is available in 32 languages, including Korean, and the various language versions are posted on their website (http://www.ralfschwarzer.de/ accessed on 10 March 2023).

Even though the GSE scale has been utilized in numerous studies, the psychometric properties of the Korean version of the GSE scale in older Korean immigrants with diabetes residing in the United States have not been investigated. For future research on self-efficacy related to self-care for older Korean immigrants with diabetes, it is crucial to examine the reliability and validity of the GSE scale Korean version. The findings of this study can also be applied to future studies on self-care for other Korean populations. The GSE scale Korean version translated and validated by Lee et al. [12] was utilized in this study. The GSE scale measures the general level of self-efficacy to deal with day-to-day challenges and stressful life events. One of the questions is "If I try hard enough, I can always solve difficult problems". The GSE scale consists of 10 items. Each item contains four possible responses: "not at all" (1 point), "barely true" (2 points), "moderately true" (3 points), and "exactly true" (4 points). The total score is the sum of all items. The total score ranges between 10 and 40, with higher scores indicating greater self-efficacy.

## 1.1. Background

Among the various immigrant groups in the United States, Korean immigrants are one of the ethnic minorities. According to Migration Information Source [13], 16% of the total Korean immigrant population is constituted of those over 65 years of age, which is slightly higher than the overall proportion of older Americans who are immigrants (14%) [13]. The majority of older Korean immigrants are monolingual, and more than 70 percent of them have trouble comprehending medical terms and utilizing translated informational materials [14]. In contrast to younger Korean immigrants, older Korean immigrants face greater difficulties in managing diabetes due to limited English literacy and limited access to health care services [15]. They are marginalized in access to insurance and adequate treatment [16], and they have more challenges in performing self-care activities to manage their diabetes. To activate daily self-care activities, they should have confidence or belief that they can accomplish required tasks. The belief to accomplish is called self-efficacy, and measuring self-efficacy is crucial for health care professionals to provide optimal care services.

Self-efficacy measuring instruments are classified generically into general and specialized scale categories. Several instruments, including the Diabetes Management Self-Efficacy Scale (DMSES) by Lee et al. [17], evaluate self-efficacy in relation to particular behaviors or situations. Alternatively, some instruments view self-efficacy as a more general trait; the GSE scale defines self-efficacy as a person's overall competence to perform across a variety of life issues [18]. Despite the fact that a number of studies [19,20] have confirmed

that the GSE scale has a high level of construct validity, additional research is necessary for various populations.

### 1.2. Conceptual Framework

This study was guided by Orem's self-care deficit nursing theory (SCDNT) [21]. The Orem's SCDNT has been widely implemented in clinical practice [22–24]. Orem's self-care framework includes six fundamental concepts: self-care, self-care agency, therapeutic self-care demand, self-care deficit, nursing agency, and nursing system [21]. One of the six concepts, self-care agency, refers to the capacity to perform self-care, and the key concept of this study, self-efficacy, is aligned with self-care agency.

The aims of this study were to assess the reliability and validity of the Korean version of the General Self-Efficacy (GSE) scale and examine the relationships between self-efficacy and socio-demographic characteristics of older Korean immigrants with diabetes.

## 2. Materials and Methods

### 2.1. Participants

This was a cross-sectional study. Participants were recruited from two Southern California congregations serving the Korean community and from social media websites. A convenience sampling strategy was used to select participants. There are various definitions of "older adults" because the aging process is not uniform across the population because of genetic, lifestyle, and health differences [25]. In this study, an older adult is defined as a person aged 55 or older, given that the California Department of Aging provides retirement community accommodation to adults aged 55 and older [26]. The eligible participants were as follows:

- Korean immigrants who are 55 years old or older and reside in the US;
- Diagnosed with diabetes;
- Able to read and write in Korean;
- Able to give consent to participate in the survey; and
- Complete all items of the survey.

This research was approved by the Institutional Review Board of University (IRB ID number: 20-342). Before participation, the primary purpose, benefits, risks, and confidentiality rights of this study were explained to all participants. There was no compensation to participate in the research.

### 2.2. Data Collection

This research included both a paper survey and an online survey. In Southern California, a paper survey was conducted at two Korean community-based congregations. The primary researcher obtained written permission from the congregations to conduct the paper survey. After obtaining permission from the sites, recruitment flyers were posted within the churches' structures. The primary investigator evaluated the eligibility of participants. In the presence of the principal researcher, eligible participants could complete the paper survey and return it directly to the researcher on-site. The online survey was conducted using the SurveyMonkey online survey platform of Momentive Global Inc. in San Mateo, California, the United States. The hyperlink to the SurveyMonkey online survey was posted on social network websites including Instagram, Facebook, Twitter, and internet community websites for Korean immigrants. Interested individuals participated in the survey immediately through the online link, and they could share the link to encourage others to participate. The compilation of data occurred between 3 October 2020 and 30 June 2021.

### 2.3. Measures

In the questionnaire, the Korean version of the GSE scale [12] was used, along with a brief socio-demographics section containing queries about gender, age, marital status, living status, educational level, employment status, annual income, health insurance, religion, years of residency in the US, and diagnosis of diabetes.

*2.4. Data Analysis*

Version 26 of the Statistical Package for the Social Sciences (SPSS) from IBM, Chicago, Illinois, the United States was utilized for data analysis. Initially, the characteristics of the participants were analyzed using descriptive statistics, percentages, and frequencies. Using means and standard deviations, the self-efficacy level was calculated. The General Self-Efficacy (GSE) scale's psychometric properties were described using Cronbach's alpha, exploratory factor analysis (EFA), and confirmatory factor analysis (CFA). The results were compared with the results of the psychometric properties to previously published studies. The relationships between self-efficacy and participant characteristics were evaluated using independent t-tests, one-way analysis of variance (ANOVA), and Pearson's correlation coefficients.

**3. Results**

*3.1. Participants' Characteristics and Self-Efficacy*

Participants' characteristics are provided in Table 1. From the paper and online survey, 603 responses were collected. Due to the COVID-19 pandemic, the online survey received the majority of responses. On the online survey, there were numerous incomplete responses. $n = 171$ was the total number of participants who met all inclusion criteria after deletion of incomplete data.

**Table 1.** Participants' Characteristics.

| Variables | Response | $n = 171$ | Percentage (%) |
|---|---|---|---|
| Gender | Male | 83 | 48.5 |
| | Female | 88 | 51.5 |
| Age (years) | 55–59 | 42 | 24.6 |
| | 60–64 | 42 | 24.6 |
| | 65–69 | 30 | 17.5 |
| | 70–74 | 14 | 8.2 |
| | 75–79 | 14 | 8.2 |
| | 80–84 | 17 | 9.9 |
| | More than 85 years | 12 | 7.0 |
| Marital status | Never married | 1 | 0.6 |
| | Married | 118 | 69.0 |
| | Separated | 4 | 2.3 |
| | Divorced | 23 | 13.5 |
| | Widowed | 25 | 14.6 |
| Living status | Living in facilities | 1 | 0.6 |
| | Living alone | 38 | 22.2 |
| | Living with family or relatives | 129 | 75.4 |
| | Living with non-family or friends | 3 | 1.8 |
| Educational level | Less than high school graduate | 13 | 7.6 |
| | High school graduate | 33 | 19.3 |
| | College or associate degree | 44 | 25.7 |
| | Bachelor's degree or higher | 81 | 47.4 |
| Employment status | Employed | 78 | 45.6 |
| | Unemployed | 93 | 54.4 |

**Table 1.** *Cont.*

| Variables | Response | *n* = 171 | Percentage (%) |
|---|---|---|---|
| Annual income | Less than 10 k | 21 | 12.3 |
| | 10 k–19,999 | 41 | 24.0 |
| | 20 k–29,999 | 12 | 7.0 |
| | 30 k–39,999 | 18 | 10.5 |
| | 40 k–49,999 | 11 | 6.4 |
| | 50 k–59,999 | 16 | 9.4 |
| | 60 k–69,999 | 9 | 5.3 |
| | 70 k–79,999 | 9 | 5.3 |
| | 80 k–89,999 | 5 | 2.9 |
| | 90 k–99,999 | 4 | 2.3 |
| | 100 k or more | 25 | 14.6 |
| Health insurance | Medicare | 45 | 26.3 |
| | Medi-Cal | 49 | 28.7 |
| | Private insurance | 60 | 35.1 |
| | Uninsured | 17 | 9.9 |
| Religion | Christianity | 144 | 84.2 |
| | Buddhist | 2 | 1.2 |
| | Islam | 0 | 0.0 |
| | Hinduism | 0 | 0.0 |
| | Other | 2 | 1.2 |
| | None | 23 | 13.4 |
| | Less than 10 years | 6 | 3.5 |
| | 10–19 | 27 | 14.6 |
| Years of residency | 20–29 | 53 | 28.7 |
| in the United States | 30–39 | 52 | 27.5 |
| | 40–49 | 39 | 19.9 |
| | More than 50 years | 12 | 5.8 |
| Diagnosis of diabetes | Yes | 171 | 100.0 |
| | No | 0 | 0 |

The percentage of female participants was marginally higher than that of male participants (51.5% versus 48.5%). Of the 171 participants, 84 (49.1%) were between the ages of 55 and 64, while 87 (50.9%) were at least 65 years old. The median age of participants was 67.3 (SD = 9.9; range, 55–93). The majority of participants were married (69.0%). 25 (14.6%) and 23 (13.5%) participants were widowed and divorced, respectively. Among the 171 participants, 129 (75.4%) lived with family or relatives, 38 (22.2%) lived alone, and three (1.8%) lived with non-family or friends.

The majority of participants (73.1%) possessed a bachelor's degree or higher. Only 13 (7.6%) participants lacked a high school diploma. Regarding employment, 54.4% of respondents were unemployed, while 45.6% were employed. In addition, the annual income of 74 (43.3%) participants was less than USD 30 k, while the annual income of 29 (26.1%) participants fell between USD 30 k and USD 50 k. A total of 68 participants (39.8%) reported an annual income in excess of USD 50 k.

Among the 171 participants, 94 (55.0%) had Medicare or Medi-Cal coverage, while 17 (9.9%) did not. The preponderance of participants (84.2%) were Christian. The majority of participants (96.5%) had lived in the United States for more than 10 years, while only six (3.5%) had lived in the country for fewer than 10 years.

Each of the ten items on the General Self-Efficacy (GSE) scale has a point value between 1 and 4. Higher scores indicate a stronger sense of self-efficacy. The mean total self-efficacy score in this study was 29.6 out of 40 (SD = 3.6, range 19–39).

*3.2. Exploratory Factor Analysis*

Initial Eigenvalues derived from exploratory factor analysis indicate that two factors (coping and confidence) explain 39.3% and 11.5% of the variance, respectively. The Varimax rotation method produced a solution containing two interpretable factors, namely coping

and confidence. Six items (Q5, Q6, Q7, Q8, Q9, Q10) account for 32.3% of the item variance with factor loadings ranging from 0.56 to 0.80, whereas four items (Q1, Q2, Q3, Q4) account for 18.5% of the item variance with factor loadings ranging from 0.40 to 0.80. Overall, coping and confidence accounted for 50.8% of the variance in the variable (see Tables 2 and 3).

**Table 2.** Factor Loadings for Varimax Orthogonal and Two-Factor Solution for the Items of the GSE Scale.

| Item | Factor Loadings | |
|---|---|---|
| | **Factor 1** | **Factor 2** |
| Factor 1: Coping ($\alpha$ = 0.83) | | |
| Q10. I can usually handle whatever comes my way. | **0.80** | 0.24 |
| Q8. When I am confronted with a problem, I can usually find several solutions. | **0.78** | 0.26 |
| Q9. If I am in trouble, I can usually think of a solution. | **0.76** | 0.17 |
| Q7. I can remain calm when facing difficulties because I can rely on my coping abilities. | **0.69** | 0.36 |
| Q6. I can solve most problems if I invest the necessary effort. | **0.62** | −0.09 |
| Q5. Thanks to my resourcefulness, I know how to handle unforeseen situations. | **0.56** | 0.22 |
| Factor 2: Confidence ($\alpha$ = 0.54) | | |
| Q2. If someone opposes me, I can find the means and ways to get what I want. | −0.05 | **0.80** |
| Q3. It is easy for me to stick to my aims and accomplish my goals. | 0.17 | **0.66** |
| Q4. I am confident that I could deal efficiently with unexpected events. | 0.28 | **0.53** |
| Q1. I can always manage to solve difficult problems if I try hard enough. | 0.36 | **0.40** |

Note. *n* = 171 and Cronbach's alpha for the entire measure is 0.81.

**Table 3.** Eigenvalues, Variance Percentages, and Cumulative Percentage for Factors in 10-Item GSE Scale.

| Factor | Eigenvalue | % Variance | Cumulative % |
|---|---|---|---|
| 1. Coping | 3.93 | 32.3 | 32.3 |
| 2. Confidence | 1.15 | 18.5 | 50.8 |

### 3.3. Reliability of General Self-Efficacy Scale-Korean Version

To examine the internal consistency, Cronbach's alpha was utilized. Cronbach's alpha for the General Self-Efficacy scale as a whole is 0.81, while the alphas for the coping and confidence subscales are 0.83 and 0.54, respectively.

### 3.4. Confirmatory Factor Analysis

The relationship between latent and observed variables of the Korean variant of the GSE scale was investigated using a confirmatory factor analysis (CFA). The LISREL® program was utilized for covariance matrix-based data analysis employing maximum likelihood estimation [27].

According to Schwarzer and colleagues [28], the GSE scale revealed one universal CFA factor. Nonetheless, the Korean version of the GSE scale revealed a two-factor model in the EFA results of this study. Consequently, this study evaluated both models, including Model A's one-factor model and Model B's two-factor model. Through conducting the CFA, it was discovered that Model B, two-factor model, showed better goodness of fit to the data compared to the Model A's one-factor model. See Table 4.

**Table 4.** Goodness-of-Fit Indices for Two Models of GSE Scale-Korean Version (*n* = 171).

| Model | *df* | $\chi^2$ | $\chi^2$/*df* Ratio | AGFI | GFI | ECVI | CFI | IFI | RMSEA | 90% CI |
|---|---|---|---|---|---|---|---|---|---|---|
| A | 35 | 86.24 | 2.46 | 0.87 | 0.91 | 0.74 | 0.89 | 0.90 | 0.093 | (0.068; 0.118) |
| B | 34 | 77.57 | 2.28 | 0.88 | 0.93 | 0.70 | 0.91 | 0.91 | 0.087 | (0.061; 0.112) |

Note. *df* = degrees of freedoms; AGFI = adjusted goodness of fit; GFI = goodness of fit; ECVI = Expected Cross-Validation Index; CFI = Comparative Fit Index; IFI = Incremental Fit Index; RMSEA = Root Mean Square Error of Approximation; CI = Confidence Interval.

However, according to the results of EFA, in the two-factor Model B, the subscale confidence's reliability is 0.54, which is unacceptable. Therefore, a one-factor solution, Model A is recommended when using the Korean version of the GSE scale.

The standardized solutions by CFA for the one-factor model, Model A are described in Figure 1.

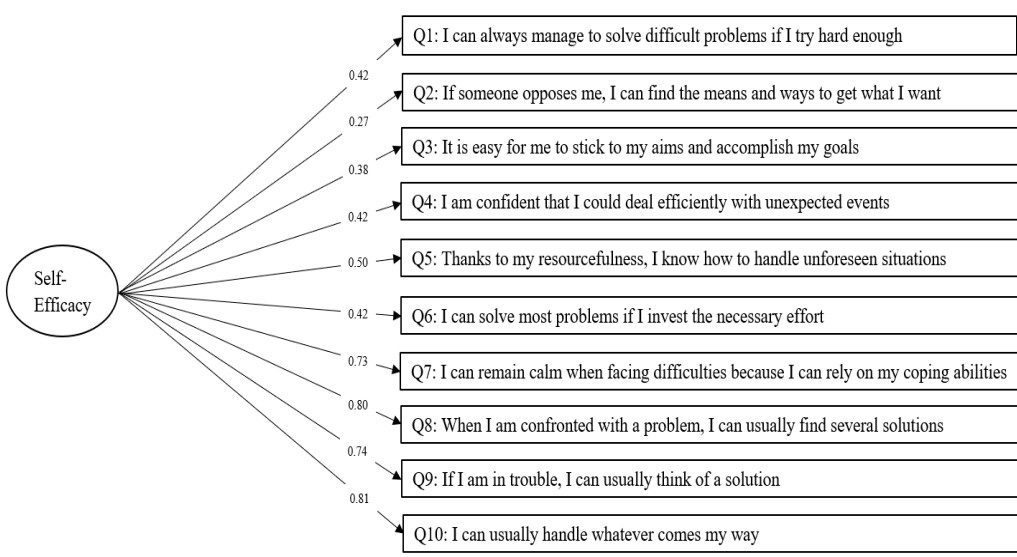

**Figure 1.** Model A: Confirmatory Factor Analysis with Standardized Solutions for a One-Factor Model with Ten Items on the General Self-Efficacy Scale, Korean Version.

### *3.5. Patient Characteristics and Self-Efficacy*

According to the correlation matrix, the higher the self-efficacy of participants, the more likely they are to have a higher educational level ($r = 0.186$, $p < 0.05$), higher annual income ($r = 0.170$, $p < 0.05$), and longer residency in the U.S. ($r = 0.248$, $p < 0.01$). The older aged participants are a lower education level ($r = -0.321$, $p < 0.01$), lower annual income ($r = -0.241$, $p < 0.01$), and longer years in the U.S. ($r = 0.239$, $p < 0.01$). See Table 5. Participants who have a higher education level are more likely to have a higher annual income ($r = 0.225$, $p < 0.01$).

**Table 5.** All Variables' Intercorrelations, Means, and Standard Deviations.

| Variables | 1 | 2 | 3 | 4 | 5 | M | SD |
|---|---|---|---|---|---|---|---|
| 1. Self-efficacy | | | | | | 29.56 | 3.60 |
| 2. Age | −0.126 | | | | | 67.29 | 9.95 |
| 3. Educational level | **0.186 *** | **−0.321 *** | | | | 3.13 | 0.98 |
| 4. Annual income | **0.170 *** | **−0.241 *** | **0.225 *** | | | 52,817.62 | 88,774.76 |
| 5. Years in the U.S. | **0.248 *** | **0.239 *** | 0.065 | 0.000 | | 30.17 | 11.88 |

Note. * Correlation is significant at the $p < 0.05$ level (two tailed). ** Correlation is significant at the $p < 0.01$ level (two tailed). Bold values indicate statistical significance.

According to the results of *t*-tests and ANOVA, living arrangement ($F(19,151) = [2.668]$, $p < 0.001$) and years of residency in the US ($F(19,151) = [2.417]$, $p = 0.002$), the two characteristics revealed statistically significant differences on the total self-efficacy score. The other characteristics, including gender, age, educational level, employment, annual income, health insurance, and religion, did not show significant differences in self-efficacy scores.

## 4. Discussion

The purpose of this study was to evaluate the psychometric properties of the Korean version of the GSE scale among older Korean immigrants with diabetes living in the United States. The results suggest that the GSE scale is legitimate and reliable for Korean

immigrants with diabetes. This study reveals that Cronbach's alpha coefficient for the overall measure is 0.81, indicating that the questionnaire was satisfactory. This study's Cronbach's alpha is comparable to 0.87 for the Thai version of the GSE among type 2 diabetes patients [29] and greater than 0.71 for the Brazil version among civil personnel [30]. The developers of GSE scale, Jerusalem and Schwarzer [11], discovered a Cronbach's alpha of 0.75. Specifically, Luszczynska and Schwarzer [31] investigated the validity of the GSE scale in numerous nations. The reliability was 0.94 among German heart disease patients, 0.89 among German cancer patients, 0.90 among Polish students, 0.87 among Polish gastrointestinal disease patients, 0.87 among Polish swimmers, and 0.86 among South Korean participants.

Significantly, the EFA revealed two factors, coping and confidence; however, Cronbach's alpha for the confidence subscale was 0.54, indicating that it was not reliable. Therefore, it is advised to use either the full GSE scale-Korean version or the subscale coping alone.

The construct validity was evaluated using exploratory and confirmatory factor analysis in this study. In contrast to previous research [11,32], the EFA revealed that the GSE consisted of two dimensions, including confidence and coping. Nonetheless, in terms of the subscale confidence showing low Cronbach's alpha, the GSE scale Korean version should be used as unidimensional. Scholz et al. [32] examined the psychometric properties of the GSE with 19,120 participants from 25 countries and demonstrated that the GSE is unidimensional. On the other hand, despite the fact that previous research has established that the GSE scale is a unidimensional and universal construct, many questions remain unanswered. Scholz et al. [32] found that Costa Ricans had the highest GSE level and Japanese had the lowest GSE sum score, indicating that the GSE sum score varied between nations. The structure of tools could also vary or change depending on the target population's culture or lifestyle. Thus, future studies need to examine other populations.

Regarding the differences in dimensions and sum score of the GSE among different nations, there could be several assumptions. First, it could be affected by different conditions of data collection. The circumstance of data collection could involve diverse uncontrolled variables to affect the results. Second, most previous studies used nonprobability sampling methods, which could be related to selection bias. Validating tools among different cultural groups is a never-ending process.

Despite being statistically significant in the correlation of the GSE and the characteristics of participants, this study showed a weak to moderate association between the GSE and the characteristics of participants. However, the correlation highlights that the positive relationship between years in the US and self-efficacy may be meaningful. The longer Korean immigrants with diabetes reside in the US, the higher their self-efficacy levels are likely to be. In addition, the higher the annual income, the higher their self-efficacy level is likely to be. It may mean that economic status and length of residency in the US affect their self-efficacy level. Additionally, the implications of the statistically significant difference between living status and self-efficacy scores should be investigated in the future.

There are no suggested cut-points for the GSE scale to distinguish between low and high self-efficacy. According to the original version, a cumulative score between 10 and 40 indicates greater self-efficacy. In this investigation, the average GSE score is 29.6 out of 40 (SD = 3.6, range 19–39), which is comparable to the 29.55 (SD = 5.32) obtained by Scholz et al. [32], who analyzed 19,120 individuals from 25 countries. In addition, the mean score of 29.6 is higher than that of Qiu et al. [33], who investigated the relationship between self-efficacy and diabetes knowledge among Chinese adult patients with diabetes. It is also greater than the 25.6 reported by Long et al. [34], who examined the role of self-efficacy as a mediator between perceived stress and quality of life among rural Chinese female patients with a history of gestational diabetes.

This study has a number of limitations. Initially, a paper questionnaire was intended to be used to collect a large sample for this research. Nonetheless, the COVID-19 pandemic occurred during the data collection phase, and the vast majority of data were collected

through an online survey. Consequently, older respondents who had trouble accessing the internet or were unfamiliar with using a computer were unable to participate in the online survey. Therefore, the results cannot be generalized to other populations. In addition, the COVID-19 pandemic may have affected the participants' perceptions of self-efficacy; however, the contextual issue was not investigated in this study. Future research must investigate the influence or causal effect of the COVID-19 pandemic on the perception of self-efficacy. This study investigated the associations between socio-demographic factors and self-efficacy. However, the relationships between self-efficacy and other constructs such as depression, anxiety, optimism were not investigated in this study. This is one of this study's limitations. Moreover, with regard to the CFA results, this study demonstrated the greatest goodness of fit in the two-factor model, despite the fact that previous research suggested one factor solution. However, since the subscale of the two-factor model, confidence's reliability, is not an acceptable value (0.54), it is recommended that the one-factor model be used. In the future, this result should be investigated.

*New Contribution to Nursing Practice*

Despite its limitations, this study has many positive qualities. This was the first study to evaluate the psychometric properties of the GSE scale among Korean diabetes immigrants in the United States, as far as we are aware. This study demonstrates that the Korean version of the GSE scale is psychometrically sound, reliable, and applicable for use with older Korean immigrants in the United States who have diabetes. The Korean version of the GSE scale can also be used to investigate how self-efficacy influences the health outcomes of diabetic patients. The Korean version of the GSE scale is probably valid for other chronic diseases, such as hypertension, in the older Korean immigrant population.

Self-care is essential in the management of diabetes, and patients must be highly motivated to engage in essential self-care behaviors. The motivation is aligned with self-efficacy, and measuring self-efficacy among patients with diabetes is essentially required.

**5. Conclusions**

This study demonstrated the validity and reliability of the Korean version of the GSE scale for measuring General Self-Efficacy in older Korean immigrants with diabetes. In addition, the Korean version of the GSE can be used to investigate factors related to self-care among Korean immigrants with other chronic conditions. The findings of this study can aid in the creation of culturally sensitive interventions and the prevention of diabetes complications.

**Author Contributions:** Conceptualization, J.E.K. and V.D.; methodology, J.E.K. and Y.-H.J.; software, J.E.K.; validation, J.E.K. and Y.-H.J.; formal analysis, J.E.K.; investigation, J.E.K.; resources, J.E.K.; data curation, J.E.K.; writing—original draft preparation, J.E.K.; writing—review and editing, J.E.K. and Y.-H.J.; visualization, J.E.K.; supervision, J.E.K. and V.D.; project administration, J.E.K.; funding acquisition, not applicable. All authors have read and agreed to the published version of the manuscript.

**Funding:** This research received no external funding.

**Institutional Review Board Statement:** Ethical review and approval were waived for this study due to exemption. This research was approved by the Institutional Review Board of Azusa Pacific University (IRB ID number: 20-342). Based on the information provided by the author, the IRB deemed this research proposal exempt from the requirements of the human subject protection regulations after reviewing this document.

**Informed Consent Statement:** Informed consent was obtained from all subjects involved in this study.

**Data Availability Statement:** The participants of this study did not give written consent for their data to be shared publicly, so due to the sensitive nature of the research, supporting data are not available.

**Conflicts of Interest:** The authors declare no conflict of interest.

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
