# Peer review of "Psychometric Properties of General Self-Efficacy (GSE) Scale Korean Version for Older Korean Immigrants with Diabetes: A Cross-Sectional Study in the United States"

_nursrep, doi:10.3390/nursrep13020074_

Round 1

Author Response

To Reviewer:

I appreciate your comments. Thank your feedback, particularly about elaborating on the manuscript and clarifying many sections for readers. My responses are in red below. Thank you again.

Reviewer 2 Report

The sentence "In 1981, Matthias Jerusalem and Ralf Schwarzer established the General Self-Efficacy (GSE) scale and validated in many countries with different patient groups." requires quoting a source from References.

What do you mean with the sentence "In addition, Korean immigrants in the United States age faster than other ethnic 65 groups"?

Revise the sentences to avoid repetition "Each item contains four possible responses. There are four possible responses:"

The sentence "This study was guided by Orem's nursing theory on self-care deficits (SCDNT)." requires quoting a source from References.

The sentence "The SCDNT of Orem has been used extensively to provide clinical guidelines for effective self-care." requires quoting some sources for the use of the SCDNT in clinical practice.

In Table 2, Cronbach's alpha for the factor Confidence (α = .54) was too loo to be used this factor. The sentence "The Cronbach's alpha for the General Self-Efficacy scale as a whole is .81, while the coping and confidence subscales have alphas of .83 and .54, respectively" also requires additional information why you have chosen to use a scale with so low internal consistence coefficient. In Table 5, you report some correlational coefficients between confidence and other variables, but the scale Confidence should not be used because of its low internal consistency. You state correctly in Discussion part "The Cronbach’s alpha of confidence is lower than .70, so it is recommended to use the entire GSE scale-Korean version or to use only the subscale coping." Why then later in Discussion you state about confidence that "However, the correlation highlights that the positive relationship between years in the US and the two subscales of confidence and coping may be meaningful."?

Why do you report the internal consistency coefficient of the total score on self-efficacy scale in your sample in the Discussion part, not in the Results part?

You state in Discussion part "The results indicate that the GSE is structurally valid and reliable for diabetic Korean immigrants." However, the internal consistency coefficient of the sub-scale Confidence is too low.

Why do you repeat in the text the same information as in Table 4? ("Through conducting the CFA, it was discovered that Model A, a one-factor model 194 did not demonstrate satisfactory goodness-of-fit based on the data (χ2 (35) = 389.98, p < 195 .01), χ2/df ratio = 11.14, AGFI = 0.64, GFI = 0.77, ECVI = 2.53, CFI = 0.68, IFI = 0.69, and 196 RMSEA = 0.244. On the other hand, the two-factor model, Model B, showed better good-197 ness-of-fit to the data (χ2 (34) = 77.57, p < .01), χ2/df ratio = 2.28, AGFI = 0.88, GFI = 0.93, 198 ECVI = 0.70, CFI = 0.91, IFI = 0.91, and RMSEA = 0.087. See Table 4.") Then, this Table 4 is redundant. Avoid repeating the same information twice! You may keep Table 4 and just comment it without repeating in text the information from this table.

Please, explain what it means when you state "According to the correlation matrix, the older aged participants are ... lower coping (r = -.168, p < .05; see Table 5)" What does lower coping mean? Unsuccessful coping? Non-effective coping? Anything else?

Introduction does not report any previous results regarding self-efficacy in people with diabetes. See, for example

Dehghan, H., Charkazi, A., Kouchaki, G. M., Zadeh, B. P., Dehghan, B. A., Matlabi, M., Mansourian, M., Qorbani, M., Safari, O., Pashaei, T., & Mehr, B. R. (2017). General self-efficacy and diabetes management self-efficacy of diabetic patients referred to diabetes clinic of Aq Qala, North of Iran. Journal of diabetes and metabolic disorders16, 8. https://doi.org/10.1186/s40200-016-0285-z

Karimy, M., Koohestani, H.R. & Araban, M. The association between attitude, self-efficacy, and social support and adherence to diabetes self-care behavior. Diabetol Metab Syndr 10, 86 (2018). https://doi.org/10.1186/s13098-018-0386-6

Lee, H., Ahn, S., & Kim, Y. (2009). Self-care, Self-efficacy, and Glycemic Control of Koreans With Diabetes Mellitus. Asian nursing research3(3), 139–146. https://doi.org/10.1016/S1976-1317(09)60025-6 

Chang, S. J., Song, M., & Im, E. O. (2014). Psychometric evaluation of the Korean version of the Diabetes Self-efficacy Scale among South Korean older adults with type 2 diabetes. Journal of clinical nursing23(15-16), 2121–2130. https://doi.org/10.1111/jocn.12133

https://www.sciencedirect.com/science/article/pii/S1976131709600256

etc.

Author Response

To Reviewer:

Thank you so much for your comments and feedback. It was helpful to clarify the study results and elaborate on the research findings. The red text below is my response to your comments. Thank you so much again.

Author Response

To Reviewer:

First, I really appreciate your insightful feedback. Your comments are helpful for improving the manuscript, particularly regarding elaborating on the results. Thank you so much for reviewing the manuscript. My responses are below (in red).

Round 2

Author Response

Dear Reviewer 1,

Thank you so much for your valuable comments. Your feedback improved the rigor of the study and have us to interpret the results with more diverse views. Really appreciate for giving your precious time to this manuscript.
